# Hemodynamic Performance of Two Current-Generation Transcatheter Heart Valve Prostheses in Severely Calcified Aortic Valve Stenosis

**DOI:** 10.3390/jcm11154570

**Published:** 2022-08-05

**Authors:** Max Potratz, Kawa Mohemed, Hazem Omran, Lasha Gortamashvili, Kai Peter Friedrichs, Werner Scholtz, Smita Scholtz, Volker Rudolph, Cornelia Piper, Tomasz Gilis-Januszewski, René Schramm, Nobuyuki Furukawa, Jan Gummert, Sabine Bleiziffer, Tanja Katharina Rudolph

**Affiliations:** 1Clinic for General and Interventional Cardiology/Angiology, Herz-und Diabeteszentrum NRW, Ruhr-Universität Bochum, 32545 Bad Oeynhausen, Germany; 2Clinic for Thoracic and Cardiovascular Surgery, Herz-und Diabeteszentrum NRW, Ruhr-Universität Bochum, 32545 Bad Oeynhausen, Germany

**Keywords:** TAVI, calcification, balloon-expandable, self-expandable

## Abstract

Background: Treatment of severely calcified aortic valve stenosis is associated with a higher rate of paravalvular leakage (PVL) and permanent pacemaker implantation (PPI). We hypothesized that the self-expanding transcatheter heart valve (THV) prostheses Evolut Pro (EPro) is comparable to the balloon-expandable Sapien 3 (S3) regarding hemodynamics, PPI, and clinical outcome in these patients. Methods: From 2014 to 2019, all patients with very severe calcification of the aortic valve who received an EPro or an S3 THV were included. Propensity score matching was utilized to create two groups of 170 patients. Results: At discharge, there was significant difference in transvalvular gradients (EPro vs. S3) (dPmean 8.1 vs. 11.1 mmHg, *p* ≤ 0.001) and indexed effective orifice area (EOAi) (1.1 vs. 0.9, *p* ≤ 0.001), as well as predicted EOAi (1 vs. 0.9, *p* ≤ 0.001). Moderate patient prosthesis mismatch (PPM) was significantly lower in the EPro group (17.7% vs. 38%, *p* ≤ 0.001), as well as severe PPM (2.9% vs. 8.8%, *p* = 0.03). PPI and the PVL rate as well as stroke, bleeding, vascular complication, and 30-day mortality were comparable. Conclusions: In patients with severely calcified aortic valves, both THVs performed similarly in terms of 30-day mortality, PPI rate, and PVL occurrence. However, patient prothesis mismatch was observed more often in the S3 group, which might be due to the intra-annular design.

## 1. Introduction

Since 2002, transcatheter aortic valve implantation (TAVI) has gradually developed from being a method reserved to treat aortic valve stenosis in high-risk patients to an alternative to surgical aortic valve replacement in intermediate and even low-risk patients [1,2,3].

However, despite its favorable outcome data, TAVI itself has been associated with severe complications [1,2,3]. Atrioventricular blockage, paravalvular leakage, stroke or vascular complication still pose a substantial problem [4].

Several risk factors have been identified as underlying causes for complications resulting in a worse outcome following TAVI. Severe calcification of the aortic valve leaflets, annulus, and left ventricular outflow tract is a particular challenging risk factor [5,6]. The degree and distribution of calcification have been linked to the permanent pacemaker implantation rate in balloon-expandable as well as self-expandable valves [7,8]. Paravalvular leakage, itself associated with a detrimental effect on prognosis [9], was found significantly more often in patients with a high calcification burden of the aortic valve [5]. The impact on stroke remains unclear, with some studies finding no correlation [10,11], while newer data reported by Spaziano et al. found a two-fold increase in the stroke rate and mortality at one year following TAVI in patients with moderate or severe left ventricular outflow tract (LVOT) calcification [6]. Furthermore, a study by Aggarwal et al. reported a significantly higher number of cranial embolization in patients with a higher valvular Agatston score [12]. The risk for annular rupture as well as mortality has been associated with valve calcification [13,14].

Addressing the procedural challenges when treating highly calcified valves as well as receiving the best hemodynamic outcomes in these patients is of great importance.

Self-expanding valves have a fundamentally different way of exerting pressure on the annulus than balloon-expandable valves [15]. A high calcification burden may impede TAVI deployment regardless of the expanding mechanism [16]. Much engineering effort has been undertaken to counter this, with additional features of modern transcatheter heart valves [17,18,19].

The Sapien 3 features a specific sealing skirt to tackle PVL for which several studies reported favorable results [18,20]. The Evolut Pro has an additional outer pericardium sheath over the bottom 1 ½ mesh segments to achieve better sealing, which translates into a reduced PVL rate [21].

The purpose of this study was to investigate the recent self-expanding THV generation—the Evolut Pro, in a head-to-head comparison against the Sapien 3 in the setting of severe calcification in propensity score-matched cohort data from our institution.

## 2. Materials and Methods

This retrospective single-center study included all patients with severe calcified aortic valve stenosis who received either an Evolut Pro (EPro, Medtronic, Minneapolis, Minnesota) or a Sapien 3 (S3, Edwards Lifesciences, Irvine, CA, USA) THV from June 2014 to December 2019. Patients were recruited from the ongoing prospective TAVI register at our institution. In total, 1988 patients were screened for enrollment in this study. All patients presented with a symptomatic, severe aortic valve stenosis as defined by ESC/AHA guidelines [22,23]. Every patient received a pre-procedural computed tomography of the heart and complete aorta. CT analysis was performed using 3 mensio structural heart software (Pie Medical Imaging, Maastricht, The Netherlands). The calcification burden was determined using a contrast-enhanced computed tomogram [24]. The region of interest was defined as reaching from the coronary ostia to the basal plane with further retrograde 10 mm into the LVOT. The severity of calcification was determined as extensive calcifications of all three cusps of the aortic valve as well as the LVOT [25]. Since exclusively an annular diameter from 18 mm to 26.4 mm can be treated with both studied THV prostheses, only patients with annular dimension within this range were included in the current analysis.

In total, 440 patients presented with severely calcified aortic valve complex and were subsequently enrolled in this analysis; 263 patients received an S3, and 177 patients received an EPro. All patients gave written informed consent for the procedure. The study protocol was approved by the local review board and ethics committee.

Eligibility of the individual candidate for TAVI had been decided by the local institutional heart team. The transfemoral artery was the preferred access route, but transsubclavian access was used when deemed more favorable in the presence of vascular calcification, small caliber, or kinking. In each group, 1 patient was treated via the subclavian access route while all others were treated transfemorally. Prosthesis selection was suggested by the heart team, but the final decision was made by the operating physician.

Echocardiographic evaluation was standardized and performed by an intervention-independent echocardiographer before TAVI and pre-discharge according to Valve Academic Research Consortium (VARC-2) recommendations [26]. The mean aortic valve gradient was measured by using continuous wave Doppler. Paravalvular leakage was classified as none/trace, mild, moderate, and severe [27]. The effective orifice area (EOA) was assessed using the continuity equation and indexed to the body surface area (EOAi). Measured EOAi was compared to predicted EOAi, which was obtained from published data [28]. PPM was classified as per the guidelines using the EOAi as follows: severe (<0.65 cm^2^/m^2^), moderate (0.65 to 0.85 cm^2^/m^2^), and not significant (>0.85 cm^2^/m^2^) in the general population, and as severe (<0.56 cm^2^/m^2^) or moderate (0.56 to 0.70 cm^2^/m^2^) in the obese population (body mass index ≥ 30 kg/m^2^) [29].

Procedural outcomes were reported according to the VARC2 consensus; 30-day mortality as well as stroke, life-threatening bleeding, stage 2 or 3 acute kidney injury, coronary artery obstruction requiring intervention, major vascular complication, and valve-related dysfunction requiring repeat procedure were assessed.

Statistical analysis: Continuous variables are expressed as the mean ± standard deviation (SD) and were compared among groups using Student’s *t*-test. Categorical data are presented as numbers and percentages. The chi-square test or Fisher’s exact test were used to evaluate differences among groups. A comparison of unpaired and paired continuous data was performed with the Wilcoxon rank sum test. Propensity score matching using the “nearest neighbor” algorithm with a caliper of 0.1 was used to match patients receiving EPro and S3 in a 1:1 fashion. The EPro group was used as the common reference. Matching parameters were the area-derived aortic annulus size, device landing zone calcification (DLZC) score, ejection fraction, age, and gender. A *p* value < 0.05 was considered significant. Statistical analysis was performed using Statistica 13.5 (TIBCO, Palo Alto, CA, USA) and R software version 3.6.3 (The R Foundation for Statistical Computing, Vienna, Austria).

## 3. Results

EPro patients were matched to S3 patients to control for confounders. Propensity score matching resulted in 170 pairs. The baseline characteristics of the unmatched and matched population are shown in Table 1. There were no differences regarding age (years): 82.5 ± 5.1 vs. 82.9 ± 6.7, *p* = 0.519, area-derived annulus size (mm^2^): 23.7 ± 1.6 vs. 23.9 ± 1.5, *p* = 0.41, gender (male): 74 (42%) vs. 71 (40%), *p* = 0.66, landing zone calcification score: 1239.4 ± 617.5 vs. 1209.3 ± 928.3, *p* = 0.73, and ejection fraction (%): 51.3 ± 7.6 vs. 51.2 ± 8.1, *p* = 0.87. The absolute standardized mean differences (d values) of all matching parameters were less than 0.1 (Figure A1), indicating adequate balance between groups and thus sufficient bias reduction [30]. Differences between groups regarding matching parameters and baseline Data are given in Table 1.

Procedural characteristics. Fluoroscopy time as well as the amount of contrast agent used was significantly higher in the EPro group compared to the S3 group (both *p* ≤ 0.001). Patients receiving the EPro experienced pre- and post-dilatation more often than those receiving an S3 THV (*p* ≤ 0.001). All data are given in Table 2.

Clinical outcomes. Analyzing for VARC 2 parameters, we found 30-day mortality, stroke rate, vascular complications (minor and major) as well as bleeding comparable between the groups. There was one case of periprocedural conversion to surgery in the S3 group because of an annular rupture. Three patients developed acute kidney failure, requiring dialysis, two in the EPro (1.2%) and one in the S3 (0.6%) group. Permanent pacemaker implantation following TAVI was (EPro vs. S3) 14.1% vs. 12.4% with a *p*-value of 0.65 (Table 2).

Hemodynamics and PPM. No differences were found regarding PVL between groups. Moderate PVL was found in 2.9% and 4.1% for EPro and S3, respectively (*p* = 0.56). Severe PVL did not occur in either group. When the calcification load of both groups was divided into thirds, no significant association with PVL was found for either THV. The calcification load between the thirds of both groups was similar; (EPro vs. S3: 1/3 698 mm^3^ vs. 470 mm^3^, 2/3 1123 mm^3^ vs. 1013 mm^3^, 3/3 1829 mm^3^ vs. 1963 mm^3^). Transthoracic echocardiography showed significant differences between groups in transvalvular gradients following implantation. Both the mean and maximal gradients were lower in the EPro group compared to the S3 group (*p* ≤ 0.001). The aortic valve area was also significantly larger in the EPro group compared to the S3 (*p* ≤ 0.001) group. EOAi was higher in the EPro group (*p* ≤ 0.001), which resulted in more moderate as well as more severe PPM in the S3 group (Table 2). Based on the predicted EOAi, PPM was generally lower, with less moderate PPM and no severe PPM in either group. However, predicted EOAi remained higher in the EPro group (*p* ≤ 0.001), and moderate PPM occurred significantly more often in the S3 group (*p* ≤ 0.001) (Table 3).

## 4. Discussion

The present study is a single-center, propensity score-matched comparison of the hemodynamic performance and clinical outcome of patients receiving the self-expanding supra-annular Evolut Pro and the balloon-expandable intra-annular Sapien 3 THV in patients with a severely calcified aortic valve. The major findings of our study are as follows: (1) Both THVs appear to be equally effective and safe in the setting of severe calcification, with a low 30-day mortality and no significant differences in PVL or PPI rate; (2) the overall achieved EOAi after implantation was significantly higher in the EPro group, with subsequent less moderate as well as severe patient prostheses mismatch.

### 4.1. Baseline Characteristics and Procedural Aspects

Overall matching was satisfactory (Appendix A, Figure A1). Atrial fibrillation was the only non-matched parameter that became insignificantly different between groups. The NYHA class and EuroSCORE II remained different between groups; however this might be due to the earlier recruitment of S3 patients starting in 2014 where TAVI was still restricted to high-risk patients, as opposed to a later enrollment of patients treated with EPro (starting in 2018), when TAVI was already considered for intermediate risk patients.

The significantly higher rate of pre- and post-dilation in self-expanding valves (SEV) in the present cohort is in line with previous comparisons of BEV and SEV [31,32]. This fact and the ability of repositioning the EPro THV may also be the reason for the significantly longer fluoroscopy time as well as the higher need for contrast agent in the EPro group. Although post-dilatation has been associated with higher cerebrovascular events [33] and contrast agent is a known risk factor for kidney injury [34], there was no significant difference in acute kidney failure as well as stroke across both groups, which is in line with recently published data comparing 224 patients with BEV (Sapien THV/XT and S 3) and SEV (CoreValve and Evolut R) [32].

### 4.2. Permanent Pacemaker Rate and Hemodynamics

The PPI rate in our study population was equally low and comparable to other published data. The reported PPI rate for the S3 covers a range from 8.5% to 20.5%, with randomized trials reporting lower rates than registries [35,36,37]. The PPI rate for the EPro ranges from 14.6% to 17.5%, although the latter stems from the FORWARD study, which also included previous generations of the Evolut THV [1,17,38]. Thus, our PPI rate of 12.4% in the BEV and 14.1% in the SEV group are in line with recent studies.

Paravalvular regurgitation has an important impact on outcome in TAVI patients. Substantial data have shown moderate and severe PVL to be associated with worse outcomes, and even mild PVL might impact prognosis [9]. In particular, a high calcification burden is a predictor for PVL [5]. We found moderate PVL in 4.1% of all patients who were treated with an S3 THV, in line with data from the PARTNER 2 (3.7%) and PARTNER 3 (3.9%) [2,35].

Moderate PVL was found in 2.9% of our patients who received an EPro THV, which is slightly higher than the rate reported in the FORWARD study (1.9%) [38] but comparable to real-world data (moderate or severe PVL: 2.8%) [17] (moderate PVL: 3.8%) [39].

A study by Mauri et al. compared BEV and SEV concerning PVL in the setting of different calcification loads of the DLZ. These authors showed that SEV had a higher risk for PVL in the setting of increased DLZ calcification [40]. In our study, we found that although BEV and SEV showed an increase in PVL at higher calcification levels, statistical significance was not reached with either SEV or BEV (Figure 1). The reason for this might be in the design improvements of the EPro compared to the Evolut R.

Transvalvular gradients were significantly lower in the supra-annular EPro THV as compared to the intra-annular S3 THV, which has been reported previously [41]. Supra-annular design has often been reported as advantageous over intra-annular design concerning PPM [31,32]. Because of its design, the S3 has been shown to be especially prone to PPM even compared to its own predecessor, the Sapien XT [42]. This resulted in a lower achieved effective orifice area and consequently higher rate of moderate as well as severe PPM in the S3 group as compared to the EPro group. While the effective orifice area is traditionally measured by the continuity equation, a novel method uses not measured, but predicted effective orifice area [28,29]. This new method generally results in a reduced occurrence of PPM and leads to a stronger association with clinical endpoints such as the remaining gradient [43]. Using predicted EOAi in our cohort, we saw a lower rate of PPM as well, but association with the remaining mean gradient was comparable using either method. Patient prosthesis mismatch as the phenomenon of an orifice area being too small in relation to the patient following surgical heart valve implantation was first described in 1978 [44]. Until this day, its impact on clinical outcome remains a matter of controversy, although two large meta-analyses of surgical patients found an association of moderate and severe PPM with all-cause and cardiac-related survival [45,46,47]. The impact of prosthesis mismatch in TAVI has not been clearly established [26,48,49]. PPM following TAVI was associated with worse survival in a Partner 1 subgroup analysis [50] as well as in a study by León del Pino et al. [51], whereas other studies found no association with mortality [48,49]. However, moderate PPM is considered a device failure according to the current valve academic research consortium (VARC) criteria [26]. It seems that PPM is associated with an impaired outcome in patients with reduced ejection fraction [52]. This has led some experts to prioritizing self-expanding THV in this patient group [53].

### 4.3. Clinical Outcomes

Life-threatening bleeding and major vascular complications in our cohort were not different between the two groups and were overall comparable to other published data [3,54].

The rate of disabling stroke was similar in both groups and comparable to other published data [17,35,36,38,39], and 30-day mortality was similar in both groups and in line with current data [17,36,38].

### 4.4. Limitations

There are several limitations to our investigation: first, it was an observational study, which makes it more prone to bias in comparison to randomized trials. Second, although we used a sophisticated matching algorithm, it cannot replace true prospective randomization. Third, it was a monocentric study, which may have been subjected to referral bias. Fourth, after matching, patients receiving an S3 THV had a slightly, but significantly, higher Euroscore II risk score, which may have affected the S3 performance evaluation in our study. Fifth, 30-day valve dysfunction was not examined in this study. Sixth, over the course of the enrollment period, the implantation strategy was adapted to the current standards, for example, from 2016, the S3 THV implantation height was increased. This may have altered the final results.

## 5. Conclusions

Overall, our study found both THVs to be safe and highly effective in patients with severely calcified aortic valve stenosis, displaying a low complication rate according to VARC II criteria. This marks a novel finding for self-expanding THVs, which seemed disadvantaged against balloon-expandable THVs in earlier generations. Patient prothesis mismatch—moderate as well as severe—was observed more often in the intra-annular balloon-expandable S3 group. Whether these findings impact long-term outcomes needs to be elucidated.

## Figures and Tables

**Figure 1 jcm-11-04570-f001:**
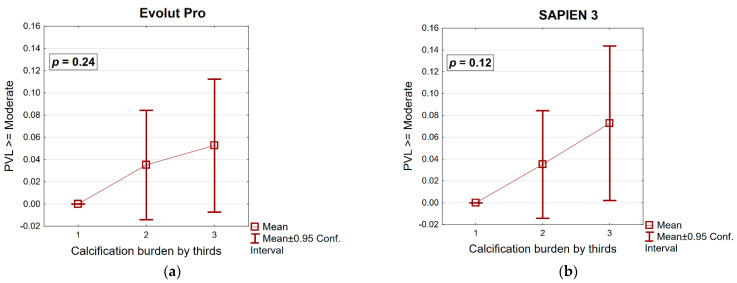
(**a**) Paravalvular leakage ≥ the medium in relationship to the calcification burden in patients that received an EPro THV. The total calcification burden of these patients was sorted by size and then divided into thirds. (**b**) Paravalvular leakage ≥ the medium in relationship to the calcification burden in patients that received an S3 THV. The calcification burden was grouped similarly.

**Table 1 jcm-11-04570-t001:** Characteristics of the matched study population grouped by patient receiving either an Evolut Pro or Sapien 3 prosthesis.

	Before Matching			After Matching		
	EPro (N = 177)	S3 (N = 263)	*p*-Value	EPro (N = 170)	S3 (N = 170)	*p*-Value
**Matching parameter**						
Age (years)	82.4 ± 5.1	81.8 ± 7.7	0.33	82.5 ± 5.1	82.9 ± 6.7	0.519
Anulus size (mm^2^)	23.85 ± 1.8	24.1 ± 1.7	0.09	23.7 ± 1.6	23.9 ± 1.5	0.41
Gender (male)	74 (41.8)	127 (48.3)	0.18	74 (42)	71 (40)	0.66
LZ Calcification score ^2^	1229.36 ± 612.5	1211.25 ± 889.8	0.81	1239.4 ± 617.5	1209.3 ± 928.3	0.73
Ejection fraction (%)	51.27 ± 7.7	51.3 ± 7.71	0.96	51.3 ± 7.6	51.2 ± 8.1	0.87
Baseline characteristics						
BMI (kg/m^2^) ^3^	26.45 ± 4.6	26.96 ± 4.9	0.27	26.4 ± 4.5	26.9 ± 4.9	0.29
EuroScore II (%)	5.25 ± 5	6.49 ± 8	0.07	5.1 ± 4.9	7 ± 9.1	0.02
NYHA	2.56 ± 0.6	2.72 ± 0.6	0.008	2.5 ± 0.6	2.7 ± 0.6	0.004
Diabetes mellitus	58 (32.8)	78 (29.7)	0.49	45 (27)	36 (21)	0.25
Arterial Hypertension	152 (85.9)	238 (90.5)	0.135	146 (86)	156 (92)	0.05
CHD ^1^	99 (55.9)	152 (57.8)	0.7	98 (57.8)	(61.2)	0.53
Atrial fibrillation	37 (20.9)	81 (30.8)	0.02	24 (13.9)	36 (21.2)	0.08
RBB	16 (9)	31 (11.8)	0.35	15 (8.9)	22 (13)	0.22
LBB	13 (7.3)	9 (3.4)	0.046	13 (7.7)	6 (3.6)	0.1
AVB I	1 (0.6)	2 (0.8)	0.85	1 (0.6)	1 (0.6)	0.99
AVB II	1 (0.6)	1 (0.4)	0.75	1 (0.6)	1 (0.6)	0.99
Previous Pacemaker	12 (6.9)	30 (11)	0.15	13 (7.6)	22 (13)	0.65
Dialysis pre TAVI	6 (3.4)	11 (4.2)	0.67	3 (1.8)	7 (4.1)	0.14
GFR (mL/min/1.72 m^2^)	57.35 ± 18.2	55.35 ± 20.7	0.3	57.7 ± 18.4	54.6 ± 20.7	0.14
Haemoglobin (mg/dL)	12.43 ± 1.7	12.46 ± 1.7	0.84	12.4 ± 1.7	12.4 ± 1.7	0.74
**Echocardiography**						
dPmean (mmHg)	49 ± 15.7	47.5 ± 17.6	0.36	49.1 ± 15.9	47.8 ± 18.1	0.48
dPmax (mmHg)	74.12 ± 22.8	74 ± 25.4	0.96	74.2 ± 23.2	74.5 ± 26.3	0.91
Aortic Valve Area (cm^2^)	0.69 ± 0.2	0.71 ± 0.2	0.41	0.69 ± 0.17	0.69 ± 0.17	0.92

Values are the mean ± SD or n (%). ^1^ CHD = coronary heart disease, ^2^ LZ = Landing zone, ^3^ BMI = Body mass index.

**Table 2 jcm-11-04570-t002:** Clinical and hemodynamic outcomes after transcatheter implantation.

	EPro (N = 170)	S3 (N = 170)	*p*-Value
Fluoroscopy time (s)	880.3 ± 355.5	747.8 ± 298.2	<0.001
Contrast agent used (mL)	120.7 ± 44.6	103.1 ± 33.4	<0.001
Prosthesis size			
23/20	1 (0.6)	0 (0)	0.32
26/23	45 (26.5)	61 (35.9)	0.06
29/26	124 (72.9)	109 (64.1)	0.08
Pre-dilatation	35 (20.6)	14 (8.3)	0.001
Post-dilatation	79 (46.7)	24 (14.3)	<0.001
Valve in Valve	0	0	
Conversion to surgery	0 (0)	1 (0.6)	0.32
Coronary obstruction	0	0	
Acute kidney failure	2 (1.2)	1 (0.6)	0.73
Myocardial infarction	0	0	
Bleeding	14 (8.4)	15 (8.8)	0.9
Minor	7 (4.2)	8 (4.7)	0.83
Major	4 (2.4)	4 (2.4)	0.97
Life threatening	3 (1.8)	3 (1.8)	0.98
Vascular complication	16 (9.6)	19 (11.2)	0.65
Minor	13 (7.8)	16 (9.4)	0.61
Major	3 (1.8)	3 (1.8)	0.98
Disabling Stroke	3 (1.8)	2 (1.2)	0.63
New PPI ^1^	24 (14.1)	21 (12.4)	0.65
30-day mortality	5 (2.9)	4 (2.4)	0.71

Values are the mean ± SD or n (%), ^1^ PPI: Permanent pacemaker implantation.

**Table 3 jcm-11-04570-t003:** Echocardiographic outcomes.

	EPro (N = 170)	S3 (N = 170)	*p*-Value
dPmean (mmHg)	8.07 ± 4.24	11.11 ± 4.14	<0.001
dPmax (mmHg)	14.09 ± 7.11	19.81 ± 6.68	<0.001
AVA post (cm^2^) ^1^	1.89 ± 0.46	1.62 ± 0.31	<0.001
Aortic regurgitation			
None/Trace	108 (63.5)	117 (68.8)	0.3
mild	57 (33.5)	46 (27.1)	0.2
moderate	5 (2.9)	7 (4.1)	0.56
severe	0 (0)	0 (0)	1
EOAi ^2^	1.07 ± 0.3	0.9 ± 0.18	<0.001
PPM moderate ^3^	30 (17.7)	65 (38)	<0.001
PPM severe	5 (2.9)	15 (8.8)	0.027
Predicted EOAi	1.05 ± 0.13	0.92 ± 0.12	<0.001
PPM moderate	1 (0.6)	25 (14.7)	<0.001
PPM severe	0 (0)	0 (0)	

Values are the mean ± SD or n (%), ^1^ AVA: Aortic valve area, ^2^ EOAi: Effective orifice area indexed, ^3^ PPM: Patient–prosthesis mismatch.

## Data Availability

The data underlying this article will be shared upon reasonable request to the corresponding author.

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
