# Peer review of "Hemodynamic Performance of Two Current-Generation Transcatheter Heart Valve Prostheses in Severely Calcified Aortic Valve Stenosis"

_jcm, 2022, doi:10.3390/jcm11154570_

Round 1

Reviewer 1 Report

Major concerns

1)    Materials and methods – I found the following sentence “This retrospective single center study included all patients who received either an Evolut Pro (EPro, Medtronic, Minneapolis, Minnesota) or a Sapien 3 (S3, Edwards Lifesciences, Irvine, California) THV from June 2014 to December 2019” not completely reliable, since the main aim of the study was to enroll only patients affected by severe calcified aortic valve stenosis. So, I suggest to report how many patients were excluded from the analysis due to the presence of only mild or moderate calcifications.

2)    Materials and methods – Since the main purpose of the study is to assess the THVs performance in severe calcified aortic valve stenosis, it is not clear (despite the proposed reference, number 24) the precise definition to define when a valve was judged severely calcified (the role of annulus and LVOT calcification?). I suggest to clear state the methods used. 

3)    Materials and methods – “Paravalvular leakage was classified as none/trace, mild, moderate and severe”. Please specify the methods adopted.

4)    Results - Baseline electrocardiographic characteristics are missing. This is a major bias of the study, since a hypothetical difference in terms of conduction disturbances can be present at baseline.

5)    Results - After propensity score matching, S3 patients presented with a significantly higher EUROSCORE II. This should be reported as a study bias, since these patients can be consider more fragile.

6)    Table 2 is too long. Procedural, in-hospital and 30-day clinical outcomes should be separated.

7)    No data have been provided concerning 30-day valve dysfunction

Minor concerns:

1)    Abstract, Result: specify the time at which results are presented

2)    Abstract, Conclusion: replace “intravalvular” with “intra-annular”

3)    Spell-out all the acronyms used the first time in the text.

4)    Introduction: replace “aortic stenosis” with “aortic valve stenosis”

5)    Introduction: “Severe calcification of the aortic valve and 38 left ventricular outflow tract is a particular challenging risk factor.” What about annular calcification?

6)    Introduction: “Much engineering effort has been undertaken to counter this, with additional features of modern transcatheter heart valves. [17,18]”. I suggest to add also this new evidence “Buono A, Gorla R, Ielasi A, Costa G, Cozzi O, Ancona M, Soriano F, De Carlo M, Ferrara E, Giannini F, Massussi M, Fovino LN, Pero G, Bettari L, Acerbi E, Messina A, Sgroi C, Pellicano M, Sun J, Gallo F, Franchina AG, Bruno F, Nerla R, Saccocci M, Villa E, D'Ascenzo F, Conrotto F, Cuccia C, Tarantini G, Fiorina C, Castriota F, Poli A, Petronio AS, Oreglia J, Montorfano M, Regazzoli D, Reimers B, Tamburino C, Tespili M, Bedogni F, Barbanti M, Maffeo D; ITAL-neo Investigators. Transcatheter Aortic Valve Replacement With Self-Expanding ACURATE neo2: Postprocedural Hemodynamic and Short-Term Clinical Outcomes. JACC Cardiovasc Interv. 2022 Jun 13;15(11):1101-1110. doi: 10.1016/j.jcin.2022.02.027. Epub 2022 May 17. PMID: 35595675.”

7)    Introduction: “The purpose of this study was to investigate the most recent self-expanding THV generation - the Evolut Pro, in a head-to-head comparison against the Sapien 3 in the setting of severe calcification in propensity score matched cohort data from our institution.” I would invite the Authors to re-word this sentence, since Evolut Pro and Sapien 3 THVs are not the most recent model anymore, because on the market are available the new iterations (respectively Evolut Pro+ and Sapien 3 Ultra).  

8)    Materials and methods –  “All patients presented with a high-grade aortic stenosis as defined by ESC/AHA guidelines”. The definition of high-grade aortic stenosis is not clear. Please replace with severe aortic valve stenosis. Moreover, were all the patients symptomatic?

Author Response

Please find the updated Text attached.

Major concerns:

1)    Materials and methods – I found the following sentence “This retrospective single center study included all patients who received either an Evolut Pro (EPro, Medtronic, Minneapolis, Minnesota) or a Sapien 3 (S3, Edwards Lifesciences, Irvine, California) THV from June 2014 to December 2019” not completely reliable, since the main aim of the study was to enroll only patients affected by severe calcified aortic valve stenosis. So, I suggest to report how many patients were excluded from the analysis due to the presence of only mild or moderate calcifications.

Response: We thank the Reviewer for the important concern. We have included the patients that were screened for the trial and furthermore restructured the paragraph for better intelligibility.

2) Materials and methods – Since the main purpose of the study is to assess the THVs performance in severe calcified aortic valve stenosis, it is not clear (despite the proposed reference, number 24) the precise definition to define when a valve was judged severely calcified (the role of annulus and LVOT calcification?). I suggest to clear state the methods used. 

Response: We thank the Reviewer for the important concern. We have restructured the paragraph for further clarification. Of all the patients screened we selected those with the most calcified aortic complex, that included calcification of the Annulus, the leaflets, as well as the LVOT. To assess calcification load we used a contrast enhanced computed tomogram as described in Source 24.

3)    Materials and methods – “Paravalvular leakage was classified as none/trace, mild, moderate and severe”. Please specify the methods adopted.

Response: We thank the Reviewer for the important concern. We now have provided a source in which our methods to grade PVL are described.

4)    Results - Baseline electrocardiographic characteristics are missing. This is a major bias of the study, since a hypothetical difference in terms of conduction disturbances can be present at baseline.

Response:  We thank the Reviewer for the important concern. We have included baseline electrocardiographic data for the unmatched and matched cohorts.

5)    Results - After propensity score matching, S3 patients presented with a significantly higher EUROSCORE II. This should be reported as a study bias, since these patients can be consider more fragile.

Response:  We thank the Reviewer for the important concern. We have included this limitation in our Paper.

6)    Table 2 is too long. Procedural, in-hospital and 30-day clinical outcomes should be separated.

Response:  We thank the Reviewer for the important concern. Splitting the table sure does improve readability. We have split the table as recommended.

7)    No data have been provided concerning 30-day valve dysfunction

Response: We thank the Reviewer for the important concern. We don't have this data and subsequently have included this shortcoming in our limitations.

Minor concerns:

1)    Abstract, Result: specify the time at which results are presented

Response:  We thank the Reviewer for the important concern. We have included the time of presentation in our paper.

2)    Abstract, Conclusion: replace “intravalvular” with “intra-annular”

Response:  We thank the Reviewer for the important concern. We have made the recommended replacement

3)    Spell-out all the acronyms used the first time in the text.

Response: We thank the Reviewer for the important concern. We have made the recommended correction.

4)    Introduction: replace “aortic stenosis” with “aortic valve stenosis”

Response: We thank the Reviewer for the important concern. We have made the recommended rectification.

5)    Introduction: “Severe calcification of the aortic valve and 38 left ventricular outflow tract is a particular challenging risk factor.” What about annular calcification?

Response: We thank the Reviewer for the important concern. We have included annular calcification as an equally challenging factor.

6)    Introduction: “Much engineering effort has been undertaken to counter this, with additional features of modern transcatheter heart valves. [17,18]”. I suggest to add also this new evidence “Buono A, Gorla R, Ielasi A, Costa G, Cozzi O, Ancona M, Soriano F, De Carlo M, Ferrara E, Giannini F, Massussi M, Fovino LN, Pero G, Bettari L, Acerbi E, Messina A, Sgroi C, Pellicano M, Sun J, Gallo F, Franchina AG, Bruno F, Nerla R, Saccocci M, Villa E, D'Ascenzo F, Conrotto F, Cuccia C, Tarantini G, Fiorina C, Castriota F, Poli A, Petronio AS, Oreglia J, Montorfano M, Regazzoli D, Reimers B, Tamburino C, Tespili M, Bedogni F, Barbanti M, Maffeo D; ITAL-neo Investigators. Transcatheter Aortic Valve Replacement With Self-Expanding ACURATE neo2: Postprocedural Hemodynamic and Short-Term Clinical Outcomes. JACC Cardiovasc Interv. 2022 Jun 13;15(11):1101-1110. doi: 10.1016/j.jcin.2022.02.027. Epub 2022 May 17. PMID: 35595675.”

Response: We thank the Reviewer for the important concern. The recommended citation has been included in the paper

7)    Introduction: “The purpose of this study was to investigate the most recent self-expanding THV generation - the Evolut Pro, in a head-to-head comparison against the Sapien 3 in the setting of severe calcification in propensity score matched cohort data from our institution.” I would invite the Authors to re-word this sentence, since Evolut Pro and Sapien 3 THVs are not the most recent model anymore, because on the market are available the new iterations (respectively Evolut Pro+ and Sapien 3 Ultra).  

Response: We thank the Reviewer for the important concern. The phrasing has been adapted to the proposed form.

8)    Materials and methods –  “All patients presented with a high-grade aortic stenosis as defined by ESC/AHA guidelines”. The definition of high-grade aortic stenosis is not clear. Please replace with severe aortic valve stenosis. Moreover, were all the patients symptomatic?

Response: We thank the Reviewer for the important concern. The replacement has been made. All patients were symptomatic. This information has been included in the text.

Reviewer 2 Report

This manuscript describes the utilization of two different THVs on severe AS and their hemodynamic performance. The manuscript is well-written and the analyses are generally adequate. I have several comments about this paper:

1. As the author mentioned, the selection bias is the major defect of this study, the matching algorithm should be well-illustrated and elucidated.

2. The devices are improved to prevent potential complications in these days. The authors should provide the implantation strategy (ex, high implant vs low implant) that included major changes in the study period.

3. The IRB number should be provided for research ethics.

4. The impact of PPM should be addressed more and should include some studies in surgical valve since it is a common issue in valve procedures.

5. The short-term effect of PPM/contrast volume during the procedure should be provided in the study groups if possible.

Author Response

Please find the updated Manuscript attached.

1. As the author mentioned, the selection bias is the major defect of this study, the matching algorithm should be well-illustrated and elucidated.

Response: We thank the Reviewer for the important concern. Apart from explaining the used Algorithm (nearest neighbor), we described the caliper that was used. This Term, which describes the closeness of the individual matches, is further explained by a Graphic provided in the Appendix. Here we expanded the explanation for greater comprehensibility. 

2. The devices are improved to prevent potential complications in these days. The authors should provide the implantation strategy (ex, high implant vs low implant) that included major changes in the study period.

Response: We thank the Reviewer for the important concern. We have included this significant fact in our text and explained that our method of implanting the S3 had undergone change. We included this in the limitations of our study.

3. The IRB number should be provided for research ethics.

Response: We thank the Reviewer for the important concern. We have provided this number in our text.

4. The impact of PPM should be addressed more and should include some studies in surgical valve since it is a common issue in valve procedures.

Response: We thank the Reviewer for the important concern. We have expanded the paragraph on PPM and included surgical studies as well as more data on its role in TAVI.

5. The short-term effect of PPM/contrast volume during the procedure should be provided in the study groups if possible.

Response: We thank the Reviewer for the important concern. Unfortunately, we did not find any short-term effect of PPM in our cohort. We provided data on acute kidney failure following THV implantation.

Round 2

Reviewer 1 Report

Thanks for the replay